# Reaction-Diffusion Systems: Self-Balancing Diffusion and the Use of the Extent of Reaction as a Descriptor of Reaction Kinetics

**DOI:** 10.3390/ijms231810511

**Published:** 2022-09-10

**Authors:** Miloslav Pekař

**Affiliations:** Faculty of Chemistry, Brno University of Technology, Purkyňova 118, 612 00 Brno, Czech Republic; pekar@fch.vut.cz

**Keywords:** diffusion, extent of reaction, independent reactions, reaction rate

## Abstract

Self-balancing diffusion is a theoretical concept that restricts the introduction of extents of reactions. This concept is analyzed in detail for general mass- and molar-based balances of reaction-diffusion mixtures, in relation to non-self-balancing cases, and with respect to its practical consequences. Self-balancing is a mathematical restriction on the divergences of diffusion fluxes. Fulfilling this condition enables the proper introduction of the extents of (independent) reactions that reduce the number of independent variables in thermodynamic descriptions. A note on a recent generalization of the concept of reaction and diffusion extents is also included. Even in the case of self-balancing diffusion, such extents do not directly replace reaction rates. Concentration changes caused by reactions (not by diffusion) are properly described by rates of independent reactions, which are instantaneous descriptors. If an overall descriptor is needed, the traditional extents of reactions can be used, bearing in mind that they include diffusion-caused changes. On the other hand, rates of independent reactions integrated with respect to time provide another overall, but reaction-only-related descriptor.

## 1. Introduction

The simultaneous occurrence of chemical reactions and diffusion is a common phenomenon encountered both in engineered and natural (biochemical) processes and covered by numerous studies and books (for example, [1,2,3,4,5,6,7,8]). The proper (thermodynamic) modelling of reaction-diffusion systems is thus important in the design of technological processes as well as in the study of processes of natural origin. Reactions and diffusion cause local changes in amounts of species (concentrations); in the former case, these changes are constrained by the reaction stoichiometry. The stoichiometric links to diffusion are rarely studied. Both reactions and diffusion are of molecular origin; their theoretical description can be based either on microscopic (statistical, kinetic theory) or macroscopic (phenomenological, continuum) approaches. In this text we are concerned with the latter type. We only mention a recent work on the statistical theory of kinetics altered by diffusion, which operates with stoichiometry [9] but not in the sense of constraints as described in this work. Several reviews have recently been published overviewing the basics of a thorough mathematical and thermodynamic description of reaction-diffusion systems at the macroscopic level. We start with a brief report on them focusing particularly on the stoichiometric impacts on reaction-diffusion systems.

Datta and Vilekar [10] based their detailed treatment of diffusion driving forces on the balance of (linear) momentum, from which reaction rates were eliminated using the equation of the continuity of mixture components, i.e., their mass balances. Thus, they do not discuss the links between the reaction rate and diffusion and the impacts of stoichiometry on these linked variables. Their approach uses barycentric velocity as the reference for component diffusion velocities but is not restricted to this reference velocity. They placed a variety of diffusion driving forces into a remarkably unified framework of continuum theory.

Whitaker [11] analyzed the balance equations of continuum mechanics and thermodynamics to discuss the relationship between diffusion forces and fluxes. This work is based on the Stokesian fluid mixture model for the partial (species) stress tensor and on the caloric equation of state, which supposes that entropy and species densities are independent variables of the total specific energy of a mixture. Though the mixture is supposed to be a reacting mixture, no other links between chemical reactions and diffusion are developed. The author concludes that the gradient of the chemical potential is not justified as a (single or leading) driving force for the diffusive flux.

Morro [12] overviewed the balance equation for reacting fluid mixtures and presents the equations for diffusion fluxes. The author came to the conclusion that the driving term for diffusion is the gradient of chemical potential rescaled by temperature. The author follows a similar line in a later paper [13]. Both papers involve considerations on the constitutive equations for diffusion fluxes and an analysis of the consequences of entropic inequality (the second law of thermodynamics). No account of the impact of stoichiometry is given in either one.

Yet another contribution by the same author [14] is concerned with the same general balances and equations, but in the context of a reacting mixture of thermoelastic solid constituents. The possibility to express the rate of reaction in terms of the extent of the reaction is briefly touched upon. However, this note is restricted only to cases in which the diffusion is negligible, and no stoichiometric consequences are described.

Already several decades ago, Bowen published a detailed mathematical analysis of the effects of the permanence of atoms (mass conservation) on reaction rates—essentially an analysis of the linear algebra consequences of stoichiometry [15]. He not only showed that rates of individual independent reactions are a consequence of this linear algebra, but also derived a limitation on the use of reaction extent as a descriptor of reaction rates. His results limit this use to non-diffusing reacting mixtures only. Later, Truesdell in his comprehensive treatment on the rational thermodynamics [16] noted briefly that this restriction is too strong and showed that reaction extent can also be used for mixtures with diffusion, providing that the diffusion is self-balancing. These interesting and important results seem not to have been analyzed or applied in chemical reaction engineering or in the thermodynamic modelling of reaction-diffusion systems. This is probably because both authors present only general results, though Bowen gives an example of a mixture of different water phases. 

The aim of this work is to look at self-balancing diffusion and the applicability of the reaction extent in more detail. The self-balancing diffusion is the principal object of this study; this work shows what are practical manifestations of the general definition of the self-balancing diffusion (Truesdell gave only the definition), how it could be detected in reality, reformulated into molar description (in contrast to the original, mass-based approach), and what are the consequences of the diffusion not being self-balancing. Another aim is to note the links to a recent generalization of extents [17].

## 2. Results and Discussion

### 2.1. Self-Balancing Diffusion and the Extent of Reaction

The charm of self-balancing diffusion is that it enables the extent of reaction to be introduced properly as a descriptor of reaction kinetics, instead of the reaction rate $J^{\alpha }$ in (S14), or $r_{\alpha }$ in (40). Then, we should consider only n−h extents instead of n rates. Let us select some reference point, usually the starting point of a reaction, in which the vector ω has a (constant) value $\omega_{0}$, and define a new vector:(1)$$\xi =\omega -\omega_{0}.$$
If the diffusion is self-balancing then, according to (42), the vector $\dot{\xi }$ is in the reaction space and thus also the vector ξ lies there [16]. The latter vector can therefore be expressed in the basis $g^{p}$ of the (n−h)-dimensional reaction space [15]; [18] (p. 153):(2)$$\xi =\overset{n-h}{\underset{p=1}{\sum }}\xi_{p}g^{p}.$$
This basis is defined as follows [18] (p. 153):(3)$$g^{p}=\overset{n}{\underset{\alpha =1}{\sum }}P^{p\alpha }e_{\alpha },\;\;p=1,\ldots ,\;n-h\;$$
where, $P^{p\alpha }$ is the stoichiometric coefficient of component α in (independent) reaction p. Combining (S16), (1)–(3) we obtain
(4)$$\overset{n-h}{\underset{p=1}{\sum }}P^{p\alpha }\xi_{p}=\frac{w_{\alpha }-w_{\alpha }^{0}}{M_{\alpha }},\;\;\;\alpha =1,\ldots ,\;n.$$

Vector **ξ** is called the vector of extents of reactions because its components (coordinates) $\xi_{p}$ in the reaction space are the extents of (independent) reactions p. The condition of self-balancing diffusion (45) can be thus understood as a condition required for the introduction of reaction extents in a reaction-diffusion system [16]. This condition seems to be unknown in the area of chemical reaction engineering and thus has never been tested. Definition (1) shows that
(5)$$\dot{\xi }=\dot{\omega }$$
and n components (in the mixture space) of the vector $\dot{\omega }$ can be replaced by n−h components (in the reaction space) of the vector $\dot{\xi }$. In this way, the number of quantities necessary for the mathematical description of thermodynamics of a reaction-diffusion system is reduced. Further, if the diffusion is self-balancing, we can express the mass fractions in arbitrary reaction time from (4) as a function of extents only:(6)$$w_{\alpha }=w_{\alpha }^{0}+M_{\alpha }\overset{n-h}{\underset{p=1}{\sum }}P^{p\alpha }\xi_{p}.$$
The components (coordinates) $\xi_{p}$ are equivalent to the extents of reactions used traditionally in chemical (engineering) kinetics. The existence of extents enables the number of independent variables to be reduced. Thus, according to (4) or (6), n mass fractions $w_{\alpha }$ can be replaced only by n−h reaction extents $\xi_{p}$. Remember that the vector **ξ** lies in the n-dimensional mixture space and, in the case of the self-balancing diffusion, also in its (n−h)-dimensional subspace (the reaction space) at the same time. The question of its (and similarly located vectors) dimension is then pointless. From a practical point of view, extents of reactions express concentration changes (caused by reaction or by diffusion) in a different manner; whereas reaction rates in (S20) express actual concentration changes per unit of time, reaction extents, as defined by (1), express these changes relatively to some fixed point in time, i.e., they are not of the ‘per time’ dimension. In other words, reaction rates are a sort of differential quantity, whereas extents are a sort of integral quantity.

Generally, the time derivative of the vector of extents in (5) still embraces diffusion rates, not only reaction rates, as (42), (S19) and (5) show: $\dot{\xi }=\sigma +\omega^{+}$. When there is no diffusion (σ=0), reaction extents are equivalent to reactions rates: $\dot{\xi }=\omega^{+}$. Equation (5) does not imply identity or one-to-one proportionality between reaction rates and reaction extents. This lack of identity or one-to-one proportionality is seen if the derivatives of extents, which for self-balancing diffusion are also located in the reaction space, are expressed on the basis of this space and combined with $\dot{\xi }=\sigma +\omega^{+}$, (3) and (S16). Accordingly,
(7)$$M_{\alpha }J^{\alpha }-\mathrm{div}\rho_{\alpha }u_{\alpha }^{w}=\rho M_{\alpha }\overset{n-h}{\underset{p=1}{\sum }}P^{p\alpha }\dot{\xi_{p}}.$$
The one-to-one proportionality is achieved only in non-diffusing mixtures ($u_{\alpha }^{w}=0$) or, at least theoretically, in “divergence-less” diffusion ($\mathrm{div}\rho_{\alpha }u_{\alpha }^{w}=0$). The latter case can be illustrated by a simple one-dimensional diffusion along the x-axis, when the y- and z-components in the diffusion velocity vector are zero. Accordingly,
(8)$$\frac{\partial \left(\rho_{\alpha }u_{\alpha x}^{w}\right)}{\partial x}=0\;\Rightarrow \;\rho_{\alpha }u_{\alpha x}^{w}=const.\;\Rightarrow \;u_{\alpha x}^{w}=\frac{const.}{\rho_{\alpha }},$$
where const. can be a function of the other two space coordinates (and time). 

Note that the whole analysis was based only on two very general principles—the mass conservation and the permanence of atoms, and thus is valid for any specific reaction-diffusion system or model. Further, it shows general impacts of the reaction stoichiometry, which is closely related to the permanence of atoms, on diffusion.

### 2.2. Self-Balancing Diffusion in Practice

What does self-balancing diffusion mean and when is diffusion self-balancing? The answer is very simple, when the rank of the matrix of atomic composition (for details see [18] (p. 151)) is equal to one. In this case, there is only one basis vector $f_{1}$ with components $S_{1\alpha }$. The atomic mass of the only atomic element (or pseudoatomic substance—see the example of NO_2_ in dimerization below), $E^{1}$, can be expressed using the molar mass of any component—without any loss of generality, let us choose $M_{1}$. Then $E^{1}=M_{1}/S_{11}$ and the general condition (44) is
(9)$$M_{1}\sigma_{1}+\frac{M_{1}}{S_{11}}S_{12}\sigma_{2}+\cdots +\frac{M_{1}}{S_{11}}S_{1n}\sigma_{n}=0.$$
Multiplying (9) by $S_{11}/M_{1}$ we obtain
(10)$$S_{11}\sigma_{1}+S_{12}\sigma_{2}+\cdots +S_{1n}\sigma_{n}\equiv \sigma \cdot f_{1}=0$$
which is the self-balancing diffusion condition (45) for the case h=1 and in this case every diffusion is self-balancing. As an example, let us use the reacting mixture of NO_2_ and N_2_O_4_, which describes the dimerization of nitrogen dioxide. If NO_2_ is selected as the pseudoatomic substance (numbered as 1), the matrix of atomic composition $\left\|S_{\sigma \alpha }\right\|$ is $\left[\begin{array}{cc} 1 & 2 \end{array}\right]$. The self-balancing condition is then
(11)$$\sigma_{{\mathrm{NO}}_{2}}+2\sigma_{N_{2}O_{4}}=0.$$
Thus, the divergences of the diffusion fluxes of the individual components are combined in this condition according to the representation of the pseudoatomic substance in these components. The general condition (44) is really very close to the self-balancing condition in this simple case: $M_{{\mathrm{NO}}_{2}}\sigma_{{\mathrm{NO}}_{2}}+M_{N_{2}O_{4}}\sigma_{N_{2}O_{4}}=0$. Note that in this example there is only one independent reaction because n−h=1.

As an example of a mixture where h=2, let us select the mixture for ammonia synthesis: N_2_ (1), H_2_ (2), and NH_3_ (3). There are two atomic substances: N (1) and H (2). The compositional matrix is
$$\left[\begin{array}{ccc} 2 & 0 & 1\\ 0 & 2 & 3 \end{array}\right].$$
The two basis vectors are: $f_{1}=\left(2;0;1\right)$ and $f_{2}=\left(0;2;3\right)$. There are two self-balancing diffusion conditions:(12a)$$2\sigma_{N_{2}}+\sigma_{{\mathrm{NH}}_{3}}=0,$$
(12b)$$2\sigma_{H_{2}}+3\sigma_{{\mathrm{NH}}_{3}}=0.$$
The first condition restricts the diffusion fluxes according to the representation of nitrogen atoms in all components, whereas the second restricts the diffusion fluxes according to hydrogen atoms. Self-balancing diffusion means that the divergences of diffusion fluxes are balanced with respect to the atomic composition of corresponding components:(13a)$$\frac{2}{M_{N_{2}}}\mathrm{div}j_{N_{2}}^{w}+\frac{1}{M_{{\mathrm{NH}}_{3}}}\mathrm{div}j_{{\mathrm{NH}}_{3}}^{w}=0,$$
(13b)$$\frac{2}{M_{H_{2}}}\mathrm{div}j_{H_{2}}^{w}+\frac{3}{M_{{\mathrm{NH}}_{3}}}\mathrm{div}j_{{\mathrm{NH}}_{3}}^{w}=0.$$
The first Equation (13a) refers to the balance with respect to nitrogen, whereas the second, (13b), refers to the balance with respect to hydrogen. Both equations contain the diffusion fluxes of reactant and product.

The self-balancing condition can be combined with the general condition (44). First, let us modify the general condition:(14)$$M\cdot \sigma =-\overset{n}{\underset{\alpha =1}{\sum }}M_{\alpha }\frac{1}{\rho M_{\alpha }}\mathrm{div}\;j_{\alpha }^{w}=-\frac{1}{\rho }\overset{n}{\underset{\alpha =1}{\sum }}\mathrm{div}\;j_{\alpha }^{w}=0.$$
Consequently,
(15)$$\overset{n}{\underset{\alpha =1}{\sum }}\mathrm{div}j_{\alpha }^{w}=0.$$
The divergence of the ammonia diffusion flux can be eliminated from (13), giving
(16)$$\frac{3}{M_{N_{2}}}\mathrm{div}j_{N_{2}}^{w}=\frac{1}{M_{H_{2}}}\mathrm{div}j_{H_{2}}^{w}.$$
This equation expresses the consequences of the self-balancing condition for diffusion in terms of the diffusion fluxes of reactants. Equation (15) can then be written by eliminating, for example, the hydrogen flux divergence using (16); the result is Equation (13a). This example thus shows that conditions (14) and (13) are consistent, but not equivalent—the specific condition (13a) does not follow from the general condition (14); the former is stronger than the latter.

Equation (16) can be transformed into molar diffusion fluxes $j_{\alpha }^{\prime w}$, which have units mol m^−2^ s^−1^:(17)$$\frac{j_{\alpha }^{w}}{M_{\alpha }}=\frac{\rho_{\alpha }u_{\alpha }^{w}}{M_{\alpha }}=c_{\alpha }u_{\alpha }^{w}=j_{\alpha }^{\prime w}.$$
The result is:(18)$$3\mathrm{div}j_{N_{2}}^{\prime w}=\mathrm{div}j_{H_{2}}^{\prime w}.$$
Thus, self-balancing diffusion means the balancing of reactant molar diffusion fluxes (their divergences) as stated by (18) in this example.

Generally, the number of self-balancing conditions is equal to the number of atoms (or pseudoatomic substances) present in the reacting mixture. Each condition balances the (divergences of the) molar diffusion fluxes of all components containing the given atom γ with respect to the number of the atom in each component. Thus,
(19)$$\sigma_{1}f_{\gamma 1}+\sigma_{2}f_{\gamma 2}+\cdots +\sigma_{n}f_{\gamma n}=0$$
where $f_{\gamma \alpha }$ represents the number of atom γ (or pseudoatomic substance) in component α. It would be desirable to look at published experimental data if such diffusion was observed and in which circumstances. Equation (18) suggests that such diffusion could occur when the initial (input) reaction mixture contains reactants in stoichiometric ratio.

### 2.3. Molar-Based View

The above analysis was based on mass balances formulated in terms of densities (mass concentrations or mass fractions). In chemistry or chemical engineering, molar amounts and molar concentrations (molar fractions) are more common. However, molar balances do not enable similarly simple and clear equations to be formulated. This is because of the fact that—in contrast to mass—molar amounts are not conserved in chemical reactions. Because $\rho_{\alpha }=c_{\alpha }M_{\alpha }$, balance (36) is transformed as:(20)$$\frac{\partial c_{\alpha }}{\partial t}+\mathrm{div}c_{\alpha }v_{\alpha }=\frac{r_{\alpha }}{M_{\alpha }}=J^{\alpha }.$$
The divergence term can be modified using the barycentric velocity: (21)$$\mathrm{div}c_{\alpha }v_{\alpha }=\mathrm{div}c_{\alpha }u_{\alpha }^{w}+c_{\alpha }\mathrm{div}\;v^{w}+v^{w}.\mathrm{grad}\;c_{\alpha }.$$
Thus, instead of (40), we have
(22)$${\dot{c}}_{\alpha }=-\mathrm{div}c_{\alpha }u_{\alpha }^{w}-c_{\alpha }\mathrm{div}\;v^{w}+J^{\alpha }$$
which, in contrast to (40), also contains the (divergence of) the barycentric velocity and is not of the form of (42). This form results only when the barycentric velocity is zero (or divergence-less):(23)$${\dot{c}}_{\alpha }=-\mathrm{div}c_{\alpha }u_{\alpha }^{w}+J^{\alpha }.$$
Note that $c_{\alpha }u_{\alpha }^{w}=j_{\alpha }^{\prime w}$, the molar diffusion flux in the barycentric reference.

Another way to transform mass balance into molar balance is to apply the material derivative with respect to the corresponding component α:(24)$$\overset{`}{c_{\alpha }^{\alpha }}=\frac{\partial c_{\alpha }}{\partial t}+v_{\alpha }.\mathrm{grad}\;c_{\alpha },\;\;\;\alpha =1,\ldots ,n.$$
The result is
(25)$$\overset{`}{c_{\alpha }^{\alpha }}=-c_{\alpha }\mathrm{div}\;v_{\alpha }+J^{\alpha }$$
but does not contain diffusion velocity. It can be introduced generally with respect to an arbitrary referential velocity: $u_{\alpha }^{\mathrm{ref}}=v_{\alpha }-v^{\mathrm{ref}}$. Then
(26)$$\overset{`}{c_{\alpha }^{\alpha }}=-\mathrm{div}c_{\alpha }u_{\alpha }^{\mathrm{ref}}-c_{\alpha }\mathrm{div}\;v^{\mathrm{ref}}+J^{\alpha }$$
which, however, is of the form of (22) and not (42), unless the referential velocity is zero (or divergence-less). Finally, we need not strive to have the material derivative in molar balances; thus, we can write (20) as
(27)$$\frac{\partial c_{\alpha }}{\partial t}+\mathrm{div}c_{\alpha }u_{\alpha }^{\mathrm{ref}}+\mathrm{div}c_{\alpha }v^{\mathrm{ref}}=J^{\alpha }.$$
This again is of the form of (22), and the form of (42) can be achieved for zero referential velocity or for the divergence-less “referential molar diffusion flow” $c_{\alpha }v^{\mathrm{ref}}$.

The concept of self-balancing can be transferred to various balances using a proper definition of vectors in the general balance form of (42). Several examples are given in Table 1.

### 2.4. When Diffusion Is Not Self-Balancing

Of course, definition (1) can be used generally, but if diffusion is not self-balancing, it yields nothing special. Combining (1) with (42), and after integration, we obtain only
(28)$$\omega -\omega_{0}==\overset{t}{\underset{0}{\int }}\left(\sigma +\omega^{+}\right)\;dt,$$
i.e., the extent is just another denomination of the integral comprising diffusion and (component) reaction rates. Note that in chemistry a reaction network with specified stoichiometric coefficients is designed first, and then the extents of individual reactions in the network are defined by relationships similar to (4):(29)$$\xi_{i}=\frac{w_{\alpha }-w_{\alpha }^{0}}{P^{i\alpha }M_{\alpha }}.$$
However, in (29), $w_{\alpha }$ should refer only to the concentration of component α reacting in reaction i, which is practically indeterminable. In contrast, the technique described in this work first naturally derives a set of acceptable and independent reactions satisfying the permanence of atoms (and related linear algebra) together with their stoichiometric coefficients and only then introduces the extents by (4); this is the mathematically correct procedure. In the general case, there is probably no need to introduce the extents of reactions, their role could be played by rates of independent reactions (independent in the sense of linear algebra [15]) derived by the reported technique. Introducing these rates into balances (S14), we obtain:(30)$$\frac{{\dot{w}}_{\alpha }}{M_{\alpha }}=-\frac{1}{\rho M_{\alpha }}\mathrm{div}\rho_{\alpha }u_{\alpha }^{w}+\frac{1}{\rho }\overset{n-h}{\underset{p=1}{\sum }}P^{p\alpha }J_{p},\;\;\alpha =1,\ldots ,\;n$$
where $J_{p}$ is the rate of (independent) reaction p [18] (p. 153). Equation (30) illustrates that changes in the concentration of each component are caused by diffusion (the first term on the right hand side) and the reactions in which it takes place (the second term)—only n−h independent reactions can be considered. In contrast, balance (S14) does not directly show individual reactions and their rates. 

### 2.5. Note on Generalized Extents; Summarizing Notes

Rodrigues et al. [17] proposed a generalization of the concept of the reaction extent (and other extents in general) to distributed reaction systems (i.e., space-distributed systems with diffusion). A more detailed comparison can be found in Appendix A.

Here, we only note that the approach of Rodrigues et al. [17] can be combined with the methodology presented in this paper by a (formal) splitting of the vector ω into reaction and diffusion contributions:(31)$$\omega =\omega_{r}+\omega_{d}.$$
Defining
(32)$${\dot{\omega }}_{r}=\omega^{+},\;{\dot{\omega }}_{d}=\sigma ,$$
using (42) we obtain
(33)$${\dot{\omega }}_{r}+{\dot{\omega }}_{d}=\omega^{+}+\sigma =\dot{\omega }.$$
The equations under (32) are analogs of (S25) and (S26) taken over from [17]. They seem to bring nothing new to the methodology of this paper. The vector $\omega^{+}$ is always in the reaction space and therefore so is vector ${\dot{\omega }}_{r}$. If the vector σ is not in the reaction space, then neither is ${\dot{\omega }}_{d}$ and the situation with introducing the extent of reaction with reference to a fixed point in time ($\omega_{0}$) is the same as that without the splitting described in (31). 

On the other hand, Equation (32) can be utilized similarly as in [17] to define generalized extents, i.e., in (S25) and (S26), which is not within the scope of this work. The advantage of introducing generalized reaction ($x_{r}$) and diffusion ($x_{d}$) extents should consist of a clear indication of which concentration changes (of a component) are caused by reactions and which by diffusion. Yet, reaction-caused changes are already given by the rates of independent reactions, cf. Equation (S20) and text around Equation (30):(34)$${\dot{\omega }}_{r}\equiv \omega^{+}=\overset{n}{\underset{\alpha =1}{\sum }}\frac{J^{\alpha }}{\rho }e_{\alpha }=\sum_{\alpha =1}^{n}\left(\frac{1}{\rho }\overset{n-h}{\underset{p=1}{\sum }}P^{p\alpha }J_{p}\right)e_{\alpha }\equiv J.$$

If the reaction rates are expressed explicitly as functions of concentrations, e.g., by the mass-action law common in kinetics, a model for the diffusion flux is employed (e.g., Fick’s law), the differential equations of mass balance can be solved, concentrations at any time (and space point) can be obtained, and from them their time derivatives and the rates of independent reactions at any time can be calculated, giving the (rate of) change caused by reactions; the remaining change is the result of diffusion, as Equations (S14) or (42) also indicate: $\sigma =\dot{\omega }-\omega^{+}$. The only difference is that the reaction rates (or diffusion fluxes) show actual rates of concentration changes, whereas the extents show changes relative to some reference. Thus, units of extent do not contain time units, as already noted below (6). Concretely, the units of ξ based on (S16) are mol g^−1^, the units of $x_{r}$ are mol m^−3^, and the units of $J_{p}$ are mol m^−3^ s^−1^. Extents are thus useful when one is not satisfied with actual or instantaneous characteristics (rates) but, for a particular reason, prefers overall or integral quantities (extents). On the other hand, reaction rates can also be integrated using (34), giving an overall descriptor. Integration in (28) provides a combined (reaction plus diffusion) overall descriptor.

Note that Equation (34) enables the compact balance form (42) to be expressed in terms of reaction rates:(35)$$\sigma =\dot{\omega }-\omega^{+}=\dot{\omega }-J=\dot{\xi }-J$$
(Equation (5) was used in the last equality). Equation (35) shows explicitly that (only) when there is no diffusion, (independent) reaction rates are directly equal to the time derivatives of the extents of these reactions.

## 3. Methods and Materials

The full theoretical overview is presented in Appendix A, here only the basic facts are reproduced (for details see [15,16,18]). A list of symbols is attached at the end; here, we only stress that the composition is expressed using the mass density $\rho_{\alpha }$, which is the mass of constituent α in unit volume of the whole mixture; in chemistry it is usually called the mass or weight concentration. 

The local mass balance for constituent α is:(36)$$\frac{\partial \rho_{\alpha }}{\partial t}+\mathrm{div}\rho_{\alpha }v_{\alpha }=r_{\alpha }$$
and for the whole mixture
(37)$$\frac{\partial \rho }{\partial t}+\overset{n}{\underset{\alpha =1}{\sum }}\mathrm{div}\rho_{\alpha }v_{\alpha }=0.$$
using the barycentric velocity $v^{w}$, balance (37) is rewritten (the material derivative relative to the barycentric velocity is symbolized by a dot):(38)$$\dot{\rho }+\rho \;\mathrm{div}\;v^{w}=0.$$

The diffusion velocity $u_{\alpha }^{w}$, defined relative to the barycentric velocity:(39)$$u_{\alpha }^{w}=v_{\alpha }-v^{w},$$
is used to rewrite the component balance (36) in the following form:(40)$$\rho {\dot{w}}_{\alpha }=-\mathrm{div}\rho_{\alpha }u_{\alpha }^{w}+r_{\alpha }.$$

Defining an abstract n-dimensional vector space, called the mixture space, with a basis $e^{\alpha }$ [15] (see also [18] (pp. 151–152)), and
(41)$$M=\overset{n}{\underset{\alpha =1}{\sum }}M_{\alpha }e^{\alpha },$$
the balances (40) can be written in the compact form
(42)$$\dot{\omega }=\sigma +\omega^{+},$$
where the vectors σ and $\omega^{+}$ with units mole kg^−1^ s^−1^ are defined in Appendix A.

Due to the linear algebra of the permanence of atoms, the mixture space is divided into two orthogonal subspaces—one (of the dimension h<n) which has no special name and is denoted as *W*, the other (of the dimension n−h) which is called the reaction space [15]. This division is an outcome of linear algebra of stoichiometry. The reaction rates and, thus, also the vector $\omega^{+}$ lie in the reaction space. In contrast, the vector of molar masses is located in the subspace *W* [15] and from the orthogonality of the two subspaces it follows that
(43)$$M\cdot \omega^{+}=0.$$
From (S17), it follows that $M\cdot \dot{\omega }=0$, and combining with (42) and (43) we obtain
(44)M·σ=0.
This is a general condition of diffusion in chemically reacting mixtures, which restricts the (divergence of) diffusion fluxes (hidden in σ). 

Diffusion is called self-balancing by Truesdell [16] if the vector σ also lies in the reaction space. Then, the vector $\dot{\omega }$ is also located in the reaction space [16]. Any vector from the reaction space is perpendicular to the basis of the subspace *W*, which we denote $f_{\sigma }\;\left(\sigma =1,\ldots ,\;h\right)$. Consequently, we can formulate the following condition for diffusion to be self-balancing:(45)$$\sigma \;\cdot \;f_{\sigma }=0.$$

## 4. Conclusions

Self-balancing diffusion is a theoretical condition that enables the proper introduction of extents of (independent) reactions as an overall reference to some fixed point (the composition at this point) conforming to the linear algebra of the permanence of atoms and stoichiometry. Unfortunately, this condition seems to have been ignored in chemistry and chemical engineering. Such extents reduce the number of independent variables in the thermodynamic description of reacting mixtures by replacing concentration variables (mass fractions, e.g.,), because the number of (independent) reactions is lower than the number of components (and component reaction rates). Theoretical descriptions of reaction-diffusion systems, including mathematical modeling of their thermodynamics, can and even should use this reduced number of variables approach. This approach is mathematically sufficient to account for (concentration) changes caused by reactions and diffusion processes. On the other hand, further theoretical work is necessary to analyze the relation of this approach to the traditional one, which ignores mathematical dependences arising from stoichiometry. 

The number of conditions of self-balancing diffusion is equal to the number of atoms or pseudoatomic substances present in a reacting mixture, i.e., forming the components of the reacting mixture. Each condition balances the divergences of component diffusion fluxes with respect to the numbers of the corresponding atom (or pseudoatomic substance) in each component. 

Even in the case of self-balancing diffusion, there is no direct one-to-one proportionality between extent and reaction rates; generally, extents (their time derivatives) also encompass diffusion rates. Direct proportionality is achieved only for “divergence-less” diffusion and, of course, when there is no diffusion. Regardless of self-balancing, the linear algebra of the permanence of atoms (stoichiometry) puts a general restriction on the divergences of diffusion fluxes (Equation (44)).

The contribution of chemical reactions to concentration changes is expressed by the rates of independent reactions, which are also derived by applying linear algebra to the permanence of atoms and stoichiometry. In this sense, there is no need to introduce extents. However, extents can be useful when there is a need for some overall (integral) characteristic of concentration changes and not only for an instantaneous one. Note that the self-balancing diffusion is not any specific model of diffusion or diffusion flux. It is a general result of two general principles—mass balance and permanence of atoms. 

It is hoped that this theoretical work will stimulate experiments aimed at finding and comparing real cases in which diffusion is or is not self-balancing, i.e., experimental works inspecting the divergence of diffusion fluxes. The first step could be to analyze some existing data on a reaction-diffusion system employing and ignoring the self-balancing condition and to compare results. Another way could be to take experimentally determined parameters—rate constants and diffusion coefficients—of a reaction-diffusion system, to perform computer modelling and look at diffusion fluxes and at fulfilling the self-balancing condition.

## Figures and Tables

**Table 1 ijms-23-10511-t001:** Definitions of vectors to be used in addressing self-balanced diffusion in various balance frameworks.

Balance	$\dot{\omega }$	$\omega^{+}$	σ
(40)	$\overset{n}{\underset{\alpha =1}{\sum }}\frac{{\dot{w}}_{\alpha }}{M_{\alpha }}e_{\alpha }$	$\overset{n}{\underset{\alpha =1}{\sum }}\frac{J^{\alpha }}{\rho }e_{\alpha }$	$-\overset{n}{\underset{\alpha =1}{\sum }}\frac{1}{\rho M_{\alpha }}\left(\mathrm{div}\rho_{\alpha }u_{\alpha }^{w}\right)e_{\alpha }$
(23)	$\overset{n}{\underset{\alpha =1}{\sum }}{\dot{c}}_{\alpha }e_{\alpha }$	$\overset{n}{\underset{\alpha =1}{\sum }}J^{\alpha }e_{\alpha }$	$-\overset{n}{\underset{\alpha =1}{\sum }}\left(\mathrm{div}c_{\alpha }u_{\alpha }^{w}\right)e_{\alpha }$
(22)	“	“	$-\sum_{\alpha =1}^{n}\left(\mathrm{div}c_{\alpha }u_{\alpha }^{w}+c_{\alpha }\mathrm{div}\;v^{w}\right)e_{\alpha }$
(27)	$\overset{n}{\underset{\alpha =1}{\sum }}\frac{\partial c_{\alpha }}{\partial t}e_{\alpha }$	“	$-\overset{n}{\underset{\alpha =1}{\sum }}\left(\mathrm{div}c_{\alpha }u_{\alpha }^{\mathrm{ref}}+\mathrm{div}c_{\alpha }v^{\mathrm{ref}}\right)e_{\alpha }$

## Data Availability

Not applicable.

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
