# Peer review of "Reaction-Diffusion Systems: Self-Balancing Diffusion and the Use of the Extent of Reaction as a Descriptor of Reaction Kinetics"

_ijms, 2022, doi:10.3390/ijms231810511_

Round 1

Reviewer 1 Report

The year 1968 (with Aris and Bowen) is not the last year of of formal reaction kinetics. Just the opposite:  1972 is the start of it with the papers of Horn, Jackson, Feinberg (in the well-known journal Arch Ratl Mech Anal), and also of Volpert, and others, in Russian. General treatment of reaction-diffusion systems also has been elaborated, see Volpert, Siegel with Mincheva, etc. It is a pity that nothing has been taken into consideration in the present manuscript. For example, stoichiometric compatibility classes, their orthogonal completions, etc. have extensively been studied. 

A recent paper about reaction extent (arXiv:2111.05253) clearly shows that the implicit definition of reaction extent, also used by the present author (S25), is not useful, and can be substituted by an explicit one. (The manuscript also contains some of the references mentioned above.) The assumption that the reaction steps are independent, almost never occurs. Similarly, the number of reaction steps is usually not smaller, but much larger than that of the number of species (in the case of combustion models a five-times factor is present, it is an empirical fact.) The kind of diffusion treated also seems to be very special, but this I only guess, I cannot surely say. Furthermore, the paper is about reaction ad diffusion. In the last few decades, I have only seen a single chemical engineer who would be capable of reading, utilizing, and improving the results of this paper. One of the reasons for this fact is that a large set of examples (theoretical or real chemical) is painfully missing. The manuscript is formally OK. If you wish it to be effective and useful outside a very small circle of researchers, you should completely rewrite it.

Author Response

The year 1968 (with Aris and Bowen) is not the last year of of formal reaction kinetics. Just the opposite:  1972 is the start of it with the papers of Horn, Jackson, Feinberg (in the well-known journal Arch Ratl Mech Anal), and also of Volpert, and others, in Russian. General treatment of reaction-diffusion systems also has been elaborated, see Volpert, Siegel with Mincheva, etc. It is a pity that nothing has been taken into consideration in the present manuscript. For example, stoichiometric compatibility classes, their orthogonal completions, etc. have extensively been studied. 

The present manuscript deals only with the self-balancing diffusion and, as far as I know, this issue was described only by Truesdell and Bowen which were cited. In addition, some newer reviews on reaction-diffusion systems were briefly described in Introduction just as an introduction to the reaction-diffusion area and to note that the self-balancing diffusion has not been investigated in more details. Anyway, as recommended, I have added similar references to works by Volpert and Mincheva, Siegel (and other ones as recommended by rev. 2).

A recent paper about reaction extent (arXiv:2111.05253) clearly shows that the implicit definition of reaction extent, also used by the present author (S25), is not useful, and can be substituted by an explicit one. (The manuscript also contains some of the references mentioned above.) The assumption that the reaction steps are independent, almost never occurs. Similarly, the number of reaction steps is usually not smaller, but much larger than that of the number of species (in the case of combustion models a five-times factor is present, it is an empirical fact.) The kind of diffusion treated also seems to be very special, but this I only guess, I cannot surely say. Furthermore, the paper is about reaction ad diffusion. In the last few decades, I have only seen a single chemical engineer who would be capable of reading, utilizing, and improving the results of this paper. One of the reasons for this fact is that a large set of examples (theoretical or real chemical) is painfully missing. The manuscript is formally OK. If you wish it to be effective and useful outside a very small circle of researchers, you should completely rewrite it.

The definition used in the present manuscript was introduced by Bowen. In my opinion, it is of explicit type, it defines the extent explicitly, cf. eq. (11). Actually, it is an integral-type quantity as is the extent defined in the work arXiv:2111.05253 though not defined directly with the use of integral, a difference between the initial and actual time point. The arXiv paper works on (material) balance of a batch system, which is a specific case, whereas the present manuscript, based on Bowen’s approach, uses just general mass balance, with field quantities. Further, the arXiv paper does not consider diffusion whereas the present manuscript is focused on diffusion and on the case of the self-balancing diffusion only. The independence of reactions is an algebraic consequence of the conservation (permanence) of atoms as demonstrated by Bowen who also proved that the number of independent reactions is smaller than the number of (reacting) species. The present manuscript was focused just on more detailed investigation of the self-balancing diffusion, as Truesdell, based on Bowen’s results, gave only its definition. I agree that the self-balancing diffusion is a special and unusual concept but perhaps interesting to be checked for its real occurrence as expressed at the conclusion. I think the simple examples presented are enough to illustrate what it should mean and be looked for in real systems and I did not want to merely multiply such examples. One example shows reacting systems where diffusion is always self-balancing, the other illustrates that the self-balancing means some condition to be fulfilled by diffusion fluxes and explains its link to reaction stoichiometry which is general.

Reviewer 2 Report

Dear Author,

I find your attempt to attack the problem, I have tried to describe in the review, as quite interesting. I suggest you to take some time and look at the manuscripts on the subject. Probably you might find some reliable way to implement your constructions. I am sorry but for the moment I cannot really grasp your suggested definitions. Moreover, it was difficult for me to understand where they all are leading to.

Sincerely

Author Response

  1. Summary

The extent of reaction has been used for a long time in chemical kinetics and chemical reaction engineering. Bowen and Truesdell, whose works the present manuscript is based on and develops their results further, put this concept into the proper framework of mass conservation and conservation of atoms in chemical reactions.

  1. Strengths

Thanks for your recognition.

iii. Weaknesses

1st paragraph. The objectives are stated in the last paragraph of Introduction, particularly at its end. This part of text was extended. The present manuscript is limited to the self-balancing diffusion and there are only two references considering this topic and both are cited. Before their citations only a brief introduction to the reaction-diffusion systems is given because this is a very broad topic otherwise requiring voluminous books. Several more general references were added, anyway. The introduction further focused on selected recent works as explained at the end of its first paragraph. End of sec. 3.1, results in 3.2-3.6, and 4 have not been published (in a peer-reviewed journal) and all this is new. Practical aspects are covered in sec. 3.2-3.4; sec. 3.5 comments on links to a recent practical generalization of the concept of the reaction extent.

ad 1) The number of independent variables is reduced because smaller number of the extents replaces higher number of the (time derivations of) concentrations (mass fractions) as written below eq. (15) and eq. (16). Text was modified to stress it.

ad 2) This is not excluded by the present analysis which just operates on the mass and atoms conservation, it is thus very general and includes various specific diffusion cases; the self-balancing diffusion being among them and (only) this is the principal subject of the present manuscript as stated in Introduction.

ad 3) See ad 1), text was modified to better state it.

Last paragraph. The integration is explained in sec. 3.5, the two paragraphs above and below eq. (44). That text is not accessible in the abstract, abstract was therefore modified. The concept of independent reactions is well-known and long-used in kinetics, thermodynamics of reacting systems, chemical reaction engineering, it is based in the linear algebra of stoichiometry and for those not familiar with it, a reference to previous work was made (below eq. (40)).

Reviewer 3 Report

This article lacks a certain novelty and is therefore not suitable for publication.

Author Response

End of sec. 3.1, results in 3.2-3.6, and 4 have not been published (in a peer-reviewed journal) and all this is new.

Round 2

Reviewer 2 Report

Dear Author,

I do not see how the argument that something exists for a while and was suggested a long time ago helps to answer my questions and address the main concern of the review.

Sincerely

Author Response

Frankly, I am totally confused by the round 2 comment. I tried to respond to each item of the round 1 step by step, I made and noted corresponding modifications in my text; there was no argument on some existence only. I have checked the actual version of the submitted manuscript and all changes are as they should be.

Reviewer 3 Report

This article is acceptable in its present form

Author Response

Thanks for your endorsement.